# Glucoselipid Biosurfactant Biosynthesis Operon of *Rouxiella badensis* DSM 100043^T^: Screening, Identification, and Heterologous Expression in *Escherichia coli*

**DOI:** 10.3390/microorganisms13071664

**Published:** 2025-07-15

**Authors:** Andre Fahriz Perdana Harahap, Chantal Treinen, Leonardo Joaquim Van Zyl, Wesley Trevor Williams, Jürgen Conrad, Jens Pfannstiel, Iris Klaiber, Jakob Grether, Eric Hiller, Maliheh Vahidinasab, Elvio Henrique Benatto Perino, Lars Lilge, Anita Burger, Marla Trindade, Rudolf Hausmann

**Affiliations:** 1Department of Bioprocess Engineering (150k), Institute of Food Science and Biotechnology, University of Hohenheim, Fruwirthstr. 12, 70599 Stuttgart, Germany; andrefahrizperdana.harahap@uni-hohenheim.de (A.F.P.H.); chantal.treinen@tum.de (C.T.); jakob.grether@uni-hohenheim.de (J.G.); eric.hiller@uni-hohenheim.de (E.H.); malihe.vahidinasab@uni-hohenheim.de (M.V.); eperino@uni-hohenheim.de (E.H.B.P.); lars.lilge@uni-hohenheim.de (L.L.); 2Cellular Agriculture, TUM School of Life Sciences, Technical University of Munich, Gregor-Mendel-Str. 4,85354 Freising, Germany; 3Department of Biotechnology, Institute for Microbial Biotechnology and Metagenomics (IMBM), University of the Western Cape, Cape Town 7535, South Africa; vanzyllj@gmail.com (L.J.V.Z.); wesleywt@gmail.com (W.T.W.); alburger@uwc.ac.za (A.B.); ituffin@uwc.ac.za (M.T.); 4Department of Organic Chemistry (130b), Institute of Chemistry, University of Hohenheim, Garbenstr. 30, 70599 Stuttgart, Germany; juergen.conrad@uni-hohenheim.de; 5Core Facility Hohenheim, Mass Spectrometry Unit, University of Hohenheim, Ottilie-Zeller-Weg 2, 70599 Stuttgart, Germany; jens.pfannstiel@uni-hohenheim.de (J.P.); iris.klaiber@uni-hohenheim.de (I.K.)

**Keywords:** biosurfactant, biosynthesis, *Escherichia coli*, glucoselipids, glycolipids, recombinant, *Rouxiella badensis*, screening

## Abstract

*Rouxiella badensis* DSM 100043^T^ had been previously proven to produce a novel glucoselipid biosurfactant which has a very low critical micelle concentration (CMC) as well as very good stability against a wide range of pH, temperature, and salinity. In this study, we performed a function-based library screening from a *R. badensis* DSM 100043^T^ genome library to identify responsible genes for biosynthesis of this glucoselipid. The identified open reading frames (ORFs) were cloned into several constructs in *Escherichia coli* for gene permutation analysis and the individual products were analyzed using high-performance thin-layer chromatography (HPTLC). Products of interest from positive expression strains were purified and analyzed by liquid chromatography/electrospray ionization tandem mass spectrometry (LC-ESI-MS/MS) and nuclear magnetic resonance (NMR) for further structure elucidation. Function-based screening of 5400 clones led to the identification of an operon containing three ORFs encoding acetyltransferase GlcA (ORF1), acyltransferase GlcB (ORF2), and phosphatase/HAD GlcC (ORF3). *E. coli* pCAT2, with all three ORFs, resulted in the production of identical *R. badensis* DSM 100043^T^ glucosedilipid with Glu-C_10:0_-C_12:1_ as the main congener. ORF2-deletion strain *E. coli* pAFP1 primarily produced glucosemonolipids, with Glu-C_10:0,3OH_ and Glu-C_12:0_ as the major congeners, predominantly esterified at the C-2 position of the glucose moiety. Furthermore, fed-batch bioreactor cultivation of *E. coli* pCAT2 using glucose as the carbon source yielded a maximum glucosedilipid titer of 2.34 g/L after 25 h of fermentation, which is 55-fold higher than that produced by batch cultivation of *R. badensis* DSM 100043^T^ in the previous study.

## 1. Introduction

Glycolipids are a diverse group of bioactive molecules with potential applications in the biomedical, pharmaceutical, cosmetic, and surfactant fields [1,2]. The surfactant activity of glycolipids arises from their amphiphilic nature, as they possess both hydrophilic glycosyl groups and hydrophobic lipid residues. Microbial glycolipids are naturally produced as complex mixtures of congeners or homologues, which vary in the number of glycosyl units, the degree of acylation, the number of conjugated lipid chains, their lipid lengths, as well as the extent of unsaturation and substitution [3]. Despite their application potential, most glycolipids are commercially limited presumably due to their low yield and relatively high production costs [4]. Moreover, identification and functional characterization of the genes and/or biosynthetic gene clusters responsible for the bioproduction of most glycolipids remain largely underexploited, except for well-known glycolipids such as rhamnolipids, sophorolipids, and mannosylerythritol lipids [5,6,7]. Identification of complete biosynthesis pathways, genetic engineering and synthetic biology could offer promising solutions to the mentioned challenges in glycolipids production.

An earlier study conducted by Kügler et al. (2015) showed that *Rouxiella badensis* DSM 100043^T^, a rod-shaped non-pathogenic Gram-negative enterobacter, produced glycolipid biosurfactant during bioreactor cultivation [8]. This was indicated by the formation of excessive foam and a significant decrease in surface tension over the batch time. The species *Rouxiella badensis* was proposed by Le Flèche-Matéos et al. (2017) who performed genome shotgun sequencing and assembly of strain *R. badensis* DSM 100043^T^ followed by phenotypic characterization [9]. A following study by Harahap et al. (2025) elucidated the chemical structure of this biosurfactant as a novel glycolipid with glucose as the carbohydrate moiety and both hydroxylated C_12:1_ and C_10:0_ fatty acids as its lipid moieties by means of NMR and LC-ESI/MS [10]. This glucoselipid biosurfactant exhibited excellent surface-active properties, shown by a low CMC of 5.69 mg/L with minimum surface tension of 24.59 mN/m as well as good stability under extreme conditions, making it highly promising for potential industrial applications in the future. The last discoveries of novel microbial glucoselipids were reported decades ago, including an anionic glucoselipid with a tetrameric oxyacyl side chain produced by *Alcanivorax borkumensis*, and the β-D-glucopyranosyl 3-(3′-hydroxytetradecanoyloxy) decanoate (Rubiwettin RG1) produced by *Serratia rubidaea* [11,12]. Since these publications, there has been a notable absence of further research on microbial glucoselipid discoveries.

This current study aimed to identify genes which might be involved in the biosynthesis of the novel glucoselipid biosurfactant in *R. badensis* DSM 100043^T^. This was accomplished by functionally screening a genome fosmid library for biosurfactant activity. Subsequent Sanger sequencing of the positive clones was performed to reveal potential open reading frames (ORFs) for glucoselipid biosynthesis and their expression in *E. coli* was assessed [13]. The recombinantly produced glucoselipids were analyzed in comparison to the glucoselipids produced by *R. badensis* DSM 100043^T^ using mass spectrometry and NMR. This was intended to demonstrate the successful recombinant production of glucoselipids in *E. coli* and to gain preliminary insight into its biosynthesis pathway. Finally, *E. coli* was cultivated in a fed-batch bioreactor process to evaluate its glucoselipid production.

## 2. Materials and Methods

### 2.1. Chemicals and Bacterial Strains

All analytical-grade chemicals were primarily purchased from Carl Roth GmbH & Co. KG (Karlsruhe, Germany), unless stated otherwise. The bacterial strains and plasmids used in this study are listed in Table 1, while employed primers are provided in Appendix A. *R. badensis* DSM 100043^T^ was grown in Luria–Bertani (LB) medium (10 g/L tryptone, 5 g/L yeast extract, and 5 g/L NaCl) at 28 °C. *E. coli* EPI300 was cultivated at 37 °C in LB medium and agar (15 g/L) supplemented with 12.5 µg/mL chloramphenicol (Merck KGaA, Darmstadt, Germany) as selection marker for fosmid pCCERI and/or supplemented with 50 µg/mL kanamycin (Merck KGaA, Germany) as selection marker for pER1.3.50.2. *P. putida* MBD1 was grown at 30 °C in LB medium or agar supplemented with 30 µg/mL apramycin (Merck KGaA, Germany) for pCCERI fosmid selection. Exconjugants of *P. putida* MBD1 were grown on M9 minimal media containing 0.2% benzoate and 30 µg/mL apramycin [14]. All *E. coli* BL21(DE3) clones containing pET21a(+) derivatives were cultivated in LB medium or agar at 37 °C supplemented with 100 µg/mL ampicillin (GERBU Biotechnik GmbH, Heidelberg, Germany) and induced with 0.1 mM isopropyl β-D-1-thiogalactopyranoside (IPTG) (GERBU Biotechnik GmbH, Heidelberg, Germany). *E. coli* K12 JM109 was grown at 37 °C in LB medium supplemented with 100 µg/mL ampicillin.

### 2.2. Genomic Library Screening

#### 2.2.1. Preparation of Genomic DNA

Genomic DNA of *R. badensis* DSM 100043^T^ was extracted from the cell pellets obtained from 100 mL of 17 h fermentation cultures following protocol described by Wang et al. (1996) with modifications where a final concentration of 0.1 mg/mL proteinase K was added into the lysis buffer and pellets were incubated at 37 °C for 3 h [18]. An amount of 2 µL RNAse A (50 U/mg) was added into the mixture after 1% SDS treatment and then incubated at 70 °C for 1 h. For extraction, an equal volume of phenol:chloroform:isoamyl alcohol (25:24:1) was added, mixed, and centrifuged at 13,000× *g* and 4 °C for 10 min. The supernatant was collected and the step was repeated. An equal volume of chloroform:isoamyl alcohol (24:1) was added to the supernatant and centrifuged under the same conditions, again repeating the step. The final supernatant was mixed with 1/10 volume of 3 M sodium acetate and ice-cold 60% isopropanol, then stored at −20 °C overnight for precipitation. After centrifugation at 13,800× *g* for 10 min, precipitated DNA was air-dried and dissolved in elution buffer (QIAprep^®^ Spin Miniprep Kit, Hilden, Germany).

Genomic DNA Was further purified according to the protocol outlined by Liles et al. (2008) [19]. DNA was size-selected using a contour-clamped homogenous electric field (CHEF) gel apparatus (Bio-Rad CHEF-DR^®^ III, Hercules, CA, USA) by electrophoresing the DNA into a low melting point agarose gel (1% *w*/*v*) using the manufacturer’s recommended settings for separating fragments between 1 kb and 100 kb. Unstained DNA was excised following staining of the edge of the agarose gel with EtBr and the DNA recovered from the gel slice using agarase treatment (NEB, Ipswich, MA, USA). The recovered DNA was precipitated following standard protocols, washed using ice-cold 70% ethanol, air-dried, and suspended in an appropriate amount of TE buffer. DNA concentration was determined using a QubitTM 2.0 fluorometer (ThermoFisher Scientific Inc., Waltham, MA, USA). End-repairing of the DNA was performed using the CopyControl^TM^ Fosmid Library Production Kit (Epicentre^®^, San Diego, CA, USA) following the manufacturer’s protocol. End-repaired DNA was finally extracted twice to remove impurities [19].

#### 2.2.2. Creation of Genomic Library

The genomic library was generated by using Copy Control^TM^ Fosmid Library Production Kit (Epicentre^®^, San Diego, CA, USA) following the manufacturer’s protocol. In this study, fosmid pCCERI was used instead of using the kit’s supplied vector. The fosmid was linearized using restriction enzyme BstZ171 (NEB, Ipswich, MA, USA) followed by dephosphorylation using 1 U shrimp alkaline phosphatase, rSAP (NEB, Ipswich, MA, USA), following the manufacturer’s protocols. The dephosphorylated fosmid was finally extracted twice to remove the impurities. The prepared genomic DNA was ligated with the pCCERI fosmid vector using T4 DNA ligase (NEB, Ipswich, MA, USA) following the manufacturer’s instructions. Library packaging was performed using MaxPlax^TM^ Lambda Packaging Extract (Epicentre^®^, San Diego, CA, USA). The phage-packed library was transfected into *E. coli* EPI300 containing conjugative helper plasmid pER1.3.50.2 to allow conjugation in other hosts. All molecular biology techniques including restriction endonuclease digestion, ligation, phosphorylation, and gel electrophoresis were performed according to standard protocols.

#### 2.2.3. High-Throughput Conjugation and Screening

The phage-infected host cells were spread on Q-trays filled with 300 mL LB agar supplemented with 12.5 µg/mL chloramphenicol and 50 µg/mL kanamycin. The colonies were picked automatically using a Genetix Qpix 2 XT robotic colony picker (Molecular Devices, Queensway, UK) and transferred into 96-well microtiter plates containing 200 µL fresh LB medium supplemented with 12.5 µg/mL chloramphenicol and 50 µg/mL kanamycin. The microtiter plates were incubated at 37 °C and 250 rpm for 17 h. The clones were stamped from the 96-well microtiter plates on Q-Trays containing 300 mL LB agar supplemented with 0.2% L-arabinose, 12.5 µg/mL chloramphenicol, and 50 µg/mL kanamycin. An overnight culture of *P. putida* MBD1 was prepared in LB medium supplemented with 50 µg/mL kanamycin and incubated at 30 °C for 17 h. The overnight culture was diluted 25-fold in LB medium containing 10 mM MgCl_2_ and incubated for 2 h at 30 °C and 125 rpm. To inhibit restriction enzymes, the culture was incubated for 10 min at 42 °C. After that, 2 mL of the 2 h culture was added to 40 mL LB medium containing 10 mM MgCl_2_ and 0.02% L-arabinose. Then, 100 µL of the *P. putida* MBD1 culture was mixed with 10 µL of the donor *E. coli* EPI 300 p.ER.1.3.50.2 culture containing the genomic library into a 96-well microtiter plate. The plates were incubated overnight at 30 °C without shaking and afterwards stamped on Q-Trays with M9-benzoate-apramycin agar which is selective for *P. putida* MBD1 [16]. The plates were incubated overnight at 30 °C and were afterwards transferred from the M9-benzoate-apramycin agar to fresh Q-trays containing LB agar supplemented with 30 µg/mL apramycin. The screening was performed for both *E. coli* EPI 300 p.ER.1.3.50.2 and *P. putida* MBD1 plates. Clones were dotted on agar plates and cultured overnight at 30 °C. A mist of paraffin was sprayed on the colonies using an airbrush and biosurfactant activity was detected by the formation of halos around the biosurfactant producing colonies [16].

#### 2.2.4. Bioinformatic Analysis of the Pathway Gene Products

Positive clones showing halos were analyzed via Sanger sequencing using primer pairs, pCCERI-FVD1—pCCERI-RVS1 and pCCERI-FVD2—pCCERI-RVS2 (Appendix A), designed for the pCCERI fosmid. Sequencings were performed using ABI Prism 377 automated DNA sequencer (Central Analytical Facility, University of Stellenbosch, Stellenbosch, South Africa). The resulting sequences were aligned to the *R. badensis* DSM 100043^T^ genome (GCA_002093665) using the CLC genomics workbench v25.0 (Qiagen, Hilden, Germany) to reveal the open reading frames (ORFs) associated with the fosmid inserts.

Alphafold (ColabFold; https://github.com/sokrypton/ColabFold; accessed on 24 May 2025) was used to predict the structure for the full length sequence of all characterized C1-phosphatases from the Huang et al. (2015) dataset and used together with the bona fide structure of the Francisella tularensis (3KD3) as input to Foldtree to construct a phylogeny tree [20,21]. The tree was then annotated using TVBOT [22]. Cagecat was used to compare the gene cluster with those on the Genbank database while clinker and TBtools-II v.2.155 were used to visualize the synteny and amino acid conservation between gene clusters [23,24].

#### 2.2.5. Cloning of ORFs into the pET21a(+) System

Initially, primer pairs (Appendix A) were designed to amplify two different fragments including removal of stop codon: fragment 1 included *glcA* and *glcB*, whereas fragment 2 included additional *glcC*. The removal of stop codon was meant to add a His-tag to the C-terminal of the protein, thus allowing the reading frame to extend and include the His-tag followed by subsequent stop codon and T7 terminator. Polymerase chain reaction (PCR) was performed using fosmid 1.8 H6 as template by Phusion^®^ High-Fidelity DNA Polymerase (NEB, Ipswich, MA, USA) following the manufacturer’s protocol. Both fragments were purified from the agarose gel using 0.5 U/µL agarase (Thermo Fisher Scientific, Waltham, MA, USA) following the manufacturer’s protocol with slight changes. The amount of 100 mg molten agarose was digested with 1 U of agarase and the fragments were precipitated with sodium acetate. Centrifugation was performed at 11,000 *g* and 4 °C. The precipitated and washed fragments were dissolved in 20 µL nuclease-free water. An initial cloning step to transfer the fragments into pJET1.2/blunt (CloneJET PCR Cloning Kit; ThermoFisher Scientific Inc., Waltham, MA, USA) was performed to facilitate cloning into pET21a. Following confirmation of cloning of the amplified fragments into pJET1.2/blunt, the fragments were liberated from these plasmids using NdeI and XhoI. Vector pET21a(+) was also linearized with NdeI and XhoI and then purified from agarose gel using NucleoSpin^®^ Gel and PCR Clean-up (MACHEREY-NAGEL GmbH & Co. KG, Düren, Germany) following the manufacturer’s protocol. Purified pET21a(+) was then ligated to fragments 1 or 2 and transformed into *E. coli* BL21(DE3). Constructs were confirmed through restriction digestion and the fidelity of the amplified fragments established by Sanger sequencing by Eurofins Genomics (Ebersberg, Germany). Positive clones carrying pET21a(+) containing fragment 1 (glcAB) and fragment 2 (glcABC) will be referred to as *E. coli* pCAT1 and *E. coli* pCAT2, respectively.

### 2.3. Gene Permutation and Expression Analysis

The goal of gene permutations is typically to understand the variability and potential outcomes that can arise from different genetic combinations involving rearrangement or recombination of genes in various ways to study their effects, interactions, and glucoselipid biosynthesis. For this, five more constructs were created carrying different combinations of ORFs. For this, all primers (Appendix A) and plasmid constructs were designed and analyzed using SnapGene software (version 7.1.1, GSL Biotech, San Diego, CA, USA). PCRs were carried out in a PCR thermal cycler peqSTAR XS (VWR^TM^, Darmstadt, Germany) using Q5^®^ High-Fidelity DNA polymerase (New England Biolabs GmbH, Frankfurt am Main, Germany) according to standard protocols. PCRs for constructing pAFP4 and pAFP5 were conducted using PrimeSTAR^®^ Max DNA Polymerase (Takara Bio Inc., Shiga, Japan) as exceptions. The Monarch^®^ DNA Gel Extraction Kit and Monarch^®^ PCR & DNA Cleanup Kit (New England Biolabs GmbH, Frankfurt am Main, Germany) were then used for plasmid fragments purification according to the manufacturer’s protocol. The respective DNA fragments were joined via Gibson Assembly using the Gibson Assembly^®^ Cloning Kit (New England Biolabs GmbH, Frankfurt am Main, Germany) [25]. The resulting constructs were transformed into chemically competent *E. coli* BL21(DE3) (New England Biolabs GmbH, Frankfurt am Main, Germany) following the manufacturer’s protocol. The fidelity of all plasmid constructs was ensured by whole plasmid sequencing (Eurofins Genomics Germany GmbH). The plasmids were extracted using innuPREP Plasmid Mini Kit 2.0 by IST Innuscreen GmbH (Berlin, Germany) according to the manufacturer’s instruction.

To evaluate the permutated gene(s) expression, shake-flask cultivations of all *E. coli* strains were performed in modified mineral salt media (MSM) supplemented with 10 g/L glucose and incubated at 37 °C and 160 rpm [10]. Induction with IPTG to a final concentration of 0.1 mM was performed when the OD_600_ reached 1.0. Samples were collected after 6 h of cultivation and subjected to an emulsification assay and oil displacement test using olive oil [26,27]. The respective samples were extracted and qualitatively analyzed by HPTLC using thin-layer chromatography (TLC) silica plate 60 RP-18 F_254S_ (Merck KGaA, Germany) as well as p-anisaldehyde solution as staining agent [10].

### 2.4. Fed-Batch Bioreactor Cultivation

Fed-batch bioreactor cultivations were performed only with *E. coli* pCAT2 and pAFP1. The first preculture was carried out in a baffled shake flask with ampicillin-supplemented LB medium using an incubator shaker (New BrunswickTM Innova 44^®^R Eppendorf AG, Hamburg, Germany) at 37 °C and 120 rpm for 12 h. The second preculture was performed in chemically defined MSM as described by Riesenberg et al. (1991) with minor modification where additional (NH_4_)_2_SO_4_ with final concentration of 5 g/L was added instead of thiamin HCl [28]. The second preculture, which was supplemented with 25 g/L glucose, was inoculated with the first preculture to reach an initial OD_600_ of 0.2 and then cultured under the same conditions as the first preculture. The bioreactor cultivations were conducted using a 30 L stainless steel fermenter (ZETA GmbH, Lieboch, Austria) initially loaded with 10 L modified Riesenberg’s medium for the batch phase. The batch medium was supplemented with 25 g/L glucose as a sole carbon source and ampicillin to prevent contamination. The bioreactor was then inoculated with previously prepared second preculture to reach an initial OD_600_ of 0.2 and run at 37 °C, pH 7.0, and initial stirrer speed of 300 rpm. The pH was controlled through addition of either 4 M H_3_PO_4_ or 20% (*v*/*v*) NH_3_ solutions. The aeration rate was set to 2 L/min at the beginning and the minimum pO_2_ level was always kept at 30% by adjusting the stirrer speed up to 900 rpm and the aeration up to 22 L/min. The end of the batch phase could be identified by typical second rise online measurement of pO_2_ level as well as offline measurement of depleting glucose concentration [29]. For the fed-batch phase, 5 L glucose (50% *w*/*v*) feed solution containing 19.7 g/L MgSO_4_.7H_2_O and 0.1 mM IPTG was pumped exponentially into the bioreactor with the initial feeding rate *F*_0_ calculated based on the formula described by Hiller et al. (2024) [30]. The desired growth rate *µ* of 0.2 h^−1^ and maintenance coefficient m of 0.05 g/(g*h) were considered into the formula. Antifoam (Struktol^®^ SB590, Hamburg, Germany) was used by manual injection into the bioreactor to avoid excessive foam build-up.

Samples were collected at 2 h intervals from the bioreactor’s sampling port and then OD_600_ was measured using a spectrophotometer (Biochrom WPACO8000, Biochrom Ltd., Cambridge, UK). Samples were then centrifuged at 4816× *g* and 4 °C for 10 min (Multifuge X3R, Thermo Fisher Scientific, USA) to separate cells from the supernatant. The cell-free supernatant was used for glucose quantification using an enzymatic assay kit (R-Biopharm AG, Darmstadt, Germany) following the manufacturer’s protocol. Prior to glucoselipid quantification, cell-free supernatant was extracted twice with ethyl acetate and then HPTLC-based (CAMAG AG, Muttenz, Switzerland) glucoselipid measurement supported with WinCATS Software 1.4.7 was performed according to Harahap et al. (2025) [10]. Each sample was applied in the length of 6 mm band on TLC silica plate 60 RP-18 F_254S_ (Merck KGaA, Germany) and developed with isopropyl acetate/methanol/acetic acid (100:10:1, *v*/*v*/*v*). The TLC plate was then derivatized using diphenylamine-aniline-phosphoric acid (DPA) solution and scanned at 620 nm for glucoselipid measurements. Purified glucoselipid by *R. badensis* DSM 100043^T^ was used as standard for this quantitative measurement. For cell dry weight (CDW) calculation, 40 mL fermentation broth was collected in triplicate and centrifuged as described previously to obtain the cell pellets. The pellets were washed with saline and dried overnight in an oven at 110 °C. CDW was measured on scale and the mean correlation factor of 3.95 was applied.

### 2.5. Structure Elucidation of Produced Glucoselipids

#### 2.5.1. Glucoselipids Extraction and Purification

Prior to glucoselipid extraction, the bioreactor culture was first centrifuged as described previously and the cell-free supernatant was extracted twice with ethyl acetate according to Harahap et al. (2025) without addition of H_3_PO_4_ [10]. The organic phase was evaporated using a rotary vacuum evaporator (R-215, Büchi Labortechnik AG, Flawil, Switzerland) at 100 mbar and 40 °C to obtain the crude extract. Purification of glucoselipid produced by *E. coli* pCAT2 was performed in the same manner as that of glucoselipid produced by *R. badensis* DSM 100043^T^ as described previously by Harahap et al. (2025) using medium-pressure liquid chromatography (MPLC; SepacoreX50, Büchi, Flawil, Switzerland) and a reverse-phase C18 column (FlashPure EcoFlex, Büchi, Flawil, Switzerland) with the flowrate of the mobile phase set to 7.5 mL/min [10]. For purification of glucoselipid produced by *E. coli* pAFP1, minor modification on the gradient of acetonitrile (ACN)/water as mobile phases was carried out: 110 min 0–100% ACN and 10 min 100–100% ACN. The eluent was collected in glass tubes, each of which is referred to as a fraction. A total amount of 90 fractions were collected during MPLC purification process and pooled fractions containing glucoselipid, seen on the derivatized HPTLC plate, were then subjected to solvent evaporation using a rotary vacuum evaporator at 10 mbar and 40 °C. The dried, purified glucoselipid samples were stored at −20 °C for subsequent structural characterization using NMR spectroscopy and LC-ESI-MS/MS.

#### 2.5.2. Nuclear Magnetic Resonance (NMR) Spectroscopy

The glycolipid samples were dissolved in 600 µL methanol-*d*4 and transferred to a standard 5 mm NMR tube. 1D and 2D NMR-spectra were recorded on an Avance HD III 600 MHz spectrometer, equipped with a 5 mm BBO Prodigy cryo-probe (Bruker, Billerica, MA, USA). ^1^H and ^13^C chemical shifts were referenced to the residual solvent signal at *δ*_H/C_ 3.35 ppm/49.0 ppm. ^1^H, ^13^C, Heteronuclear Single-Quantum Coherence (HSQC), Heteronuclear Multiple Bond Correlation (HMBC), Correlation Spectroscopy (COSY), Total Correlation Spectroscopy (TOCSY), HSQCTOCSY and selective 1D-TOCSY spectra were recorded using standard Bruker pulse sequences at 298 K. The recorded NMR spectra were processed with Topspin 4.2.0 (copyright 2022, Bruker Biospin, Billerica, MA, USA) and SpinWorks 4.2.10 (Copyright 2019, K. Marat, University of Manitoba, CA, USA).

#### 2.5.3. Mass Spectrometry Characterization

The LC-ESI/MS analysis of the glycolipid was performed on a 1290 UHPLC system (Agilent, Waldbronn, Germany) coupled to a Q-Exactive Plus Orbitrap mass spectrometer equipped with a heated electrospray ionization source (HESI, Thermo Fisher Scientific, Bremen, Germany) as previously described by Harahap et al. (2025) with modifications [10]. Glycolipids separations were achieved by an ACQUITY CSH C18 column (1.7 μm, 2.1 μm × 150 mm, Waters, Eschborn, Germany). Gradient elution was carried out at a constant flow rate of 0.3 mL/min, with specific conditions for the glucoselipid samples from both *E. coli* pCAT2 and pAFP1 detailed in Table 2. The HESI source was operated in the positive and negative ion modes with a spray voltage of 4.0 kV in the positive ion mode and 3.5 kV in the negative ion mode. The ion transfer capillary temperature was set to 350 °C and the sweep gas and auxiliary pressure rates were set to 35 and 10, respectively. The S-lens RF level was set to 50%. The Q-Exactive Plus mass spectrometer was calibrated externally in the positive and negative ion modes using the manufacturer’s calibration solutions (Pierce, Thermo Fisher Scientific, Bremen, Germany). Mass spectra were acquired at a resolution of 70,000 at *m*/*z* 200 using an Automatic Gain Control (AGC) target of 3.0 × 10^6^ of and a maximum ion injection time of 100 ms. Data-dependent MS/MS spectra in the mass range of 200 to 2000 *m*/*z* were generated for the five most abundant precursor ions with a resolution of 17,500 at *m*/*z* 200 using an AGC target of 1.0 × 10^6^ and 100 ms maximum ion injection time. Xcalibur software version 4.3.73.11 and Compound Discoverer Software version 3.3 (both Thermo Fisher Scientific, San Jose, CA, USA) were used for data acquisition and data analysis. Identification and assignment of the glucoselipids were based on the precise *m*/*z* value of the precursor ions and manual inspection of the corresponding MS/MS spectra.

## 3. Results

### 3.1. Identification of Genes Responsible for Glucoselipid Biosynthesis and Delineation of Their Functional Roles

To identify the genes responsible for glucoselipid synthesis in *R. badensis* DSM 100043^T^, a genome fosmid library was constructed for screening in *P. putida*. Following conjugation to *P. putida*, a total of 5400 clones were screened resulting in four positive clones (Appendix A). The terminal ends of the fosmid insert sequences were mapped to the *R. badensis* DSM 100043^T^ genome (GCA_002093665) to delineate the genomic fragments captured in the respective clones which resulted in biosurfactant activity. This fragment represents the sequence from ~65,000 bp until ~101,000 bp on contig NZ_MRWE01000009.1. These clones revealed a shared operon consisting of three genes encoding ORFs designated as *glc*A (ORF1), *glc*B (ORF2), and *glc*C (ORF3) (Appendix A). The first ORF, encoding GlcA (WP_017492724), belongs to N-acetyltransferase (RimL; cI34333)/N-acyltransferase superfamilies (cI17182), specifically the GCN5-related N-acetyltransferases (GNAT) family (Table 3). The second ORF, encoding GlcB (WP_009635452), belongs to the 1-acyl-sn-glycerol-3-phosphate acyltransferase (Phospholipid synthase; PlsC; cI43057) and lysophospholipid acyltransferase (LPLAT; cI17185) superfamilies. The third ORF, encoding GlcC (WP_017492722), belongs to the haloacid dehalogenase-like hydrolase (HAD-like) and phosphoserine phosphatase (PSP) superfamilies (SerB; cl21460), specifically the C-1 type of HAD phosphatases. No closely related phosphatases appear to have been characterized, and the closest solved three-dimensional structure is that of a PSP of unknown function from *Francisella tularensis* (Q5NH99; 3KD3).

Although relatives of each individual ORF from the operon occur in disparate bacterial genomes, the conservation of the three-gene operon appears to be unique to certain species in the genera *Rouxiella* (25 genomes), *Pseudomonas* (46301 genomes), *Vibrio* (31462 genomes) and *Ottowia* (133 genomes) (Appendix A). The pathways in *Rouxiella* sp. are clearly unique, sharing low amino acid similarity and clustering away from those in other bacteria that contain a similar three-gene operon (Appendix A). Conservation of the three-gene operon in selected genomes suggests limited selective pressure to maintain the complete operon and perhaps the resultant product, whereas the presence of homologs of the individual genes in other genomes could suggest that related functions play a role in synthesis or modification of lipids produced by these organisms.

Analysis of the genomes of the six representative species of *Rouxiella* classified by the current Genome Taxonomy Database reveals that the pathway seems limited to *R. badensis*, *R. chamberiensis* and the unclassified *Rouxiella* sp. WC2420 (Figure 1). The genomic context within which these three genes find themselves is quite different when comparing the region between bacteria that contain the pathway (Appendix A). Expression of these genes in rhizosphere-associated bacteria as well as co-expression with genes potentially involved in pathogenesis (Phospholipase C, Type I secretion system) and plant growth promotion (auxin efflux carrier) suggests that this operon has been adapted for several different uses in these bacteria involved with host evasion/invasion or symbiosis (Appendix A). Biosurfactants are well-known to play a role in bacterial pathogenesis as well as plant growth promotion and in line with this, *R. badensis* was recently described as an emerging onion pathogen [31,32].

To delineate which genes are necessary for glucoselipid biosynthesis, we designed a series of constructs expressing different gene combinations of the pathway to assay glucoselipid production following expression in *E. coli* (Figure 2A). Initially, the growth behavior and product analyses among all expression strains were compared to *E. coli* BL21 (DE3) carrying an unmodified vector as control. All strains, including the control, exhibited similar growth behavior in shake flask cultivation, with the exception for *E. coli* pCAT1 (Appendix A). All strains, including the control, entered the stationary phase after 8 h of cultivation, reaching an average OD_600_ of approximately 10. In contrast, *E. coli* pCAT1 only achieved an OD_600_ of approximately 6. The reduced growth rate observed in *E. coli* pCAT1 indicates signs of stress, suggesting that the strain experienced a metabolic burden. It is possible that the combination of N-acetyltransferase (*glc*A) and acyltransferase (*glc*B) expression led to cellular stress due to the accumulation of new toxic intermediates or products. Additionally, the metabolites produced (e.g., free fatty acid) may have integrated into the cell membrane, potentially triggering membrane-associated stress responses [33,34]. Using emulsification and oil displacement assays, it was observed that the expression of the encoding genes in *E. coli* pCAT2 resulted in a positive phenotype (Figure 2B). Specifically, *E. coli* pCAT2 exhibited the highest values among all tested expression strains, with an emulsification unit of 138.9 EU/mL and an oil displacement diameter of 2.95 cm. In comparison, *E. coli* pAFP1 showed approximately half of these values, 68.8 EU/mL for emulsification assay and 1.45 cm for oil displacement, yet still higher than the other expression strains and the control. Since these two parameters are indirect indicators of biosurfactant activity, a comparison with the HPTLC results in Figure 2C suggests that *E. coli* pAFP1 produced biosurfactant compounds at levels approximately half of those observed in *E. coli* pCAT2 [26,27]. Based on the p-anisaldehyde-stained RP18 TLC plate in Figure 2C, expression of all three genes (pCAT2) resulted in the production of a compound indicated by a prominent band at *R_f_* 0.79, which matched the *R_f_* value of the glucosedilipid reference produced by *R. badensis* DSM 100043^T^. Furthermore, expression of *glc*A and *glc*C (pAFP1) led to the production of several compounds, indicated by streaky bands between *R_f_* 0.5 and 0.6, whereas expression of *glc*B and *glc*C (pAFP2) did not lead to the production of any glycolipids. Expression of both *glc*A and *glc*B (pCAT1) did not result in glycolipids synthesis, suggesting an absolute requirement for the phosphatase. Expressions of individual genes also showed no glycolipids synthesis.

The source of the 3-hydroxy fatty acid donors for transfer to a glucose molecule resulting in the glycolipids are likely to come from the pool of acyl-carrier proteins, as intermediates of fatty acid synthesis, or as intermediates of fatty acid degradation in the form of long-chain acyl-CoA derivatives [35]. GlcA is clearly related to the GNAT family of acetyltransferases. The closest structural match to GlcA is 7KPS, an acetyltransferase from *Pseudomonas aeruginosa* with demonstrated ability to catalyze transfer of an acetyl group to polymyxin antibiotics (Appendix A) [36]. Although the backbone is highly conserved, the acyl acceptor side of the active site geometry and surface charge differ substantially between GlcA and 7KPS (Appendix A). It does display a greater overall positive charge as seen in 4KUA, known to perform O-acetylation of chloramphenicol. Although this enzyme family usually performs transfer of acetyl groups to ε-amino groups or primary amines on a large variety of biomolecules, rare examples have been described that either catalyze O-acetylation of hydroxyl groups [37,38,39]. Notably, a GNAT-related acetyltransferase found in *Lysobacter enzymogenes* was found capable of O-acetylation of chloramphenicol using isobutyryl-CoA and isovaleryl-CoA as substrates [40]. Additionally, GNAT-related enzymes have been found to catalyze fatty acyl transfer of 3-hydroxy fatty acids to amine groups on small molecules as in the case of phaeornamide and fatty acids as in the case of *Mycobacterium tuberculosis* Rv1347c [41,42]. All these references suggest that these enzymes can display substantial substrate flexibility; however, to our knowledge, no GNAT-related enzyme has been described performing O-acylation of hydroxyl groups with a fatty acid, likely required for synthesis of glucoselipids. The alignment of GlcA with several characterized GNAT-acetyltransferases, including examples of enzymes reported to perform O-acetylation, did not yield any information that could support the substrate selection for the enzyme, due to sequence divergence.

GlcB shows greatest structural similarity to 5KYM, a 1-acyl-sn-glycerol-3-phosphate (LPA) acyltransferase from *Thermotoga maritima* (Appendix A) [43]. This enzyme class is characterized as having a narrow substrate range; however, some are known to catalyze acyl transfer to substrates other than LPA, such as ornithine lipid synthase A or sulfur-containing aminolipid synthase from *Ruegeria pomeroyi* [44,45,46]. Emerging evidence suggests a general role for these enzymes as responsible for the addition of a second fatty acyl chain to a variety of substrates. Comparison of the active site cleft geometry with that of 5KYM would suggest that GlcB is capable of accepting much bulkier substrates (Appendix A). A hydrophobic two-helix motif present in both enzymes likely leads to it being membrane associated which would agree with a role for the biosurfactant as lipid membrane augmenter or if it’s to be exported [43].

Several HAD-like sugar phosphatases capable of dephosphorylating sugar-6-phosphates have been described from *Pseudomonas fluorescens* (PFLU2693), *Thermophilus volcanium* (Q978Y6), *Bacteroides thetaiotaomicron* (Q8A2F3), *Eubacterium rectale* (D0VWU2), *Geobacillus kaustophilus* (Q5L139), *Saccharomyces cerevisiae* dog1 and dog2 (P38774, P38773), several enzymes from *E. coli* as well as an exceptional effort to capture the substrate preference for representative members for a significant portion of the superfamily [20,47,48,49,50]. This dataset includes the substrate range for the *F. tularensis* enzyme mentioned above which displays weak activity on glucose-6-phosphate (G6P) but has demonstrable activity on a wide range of phosphorylated substrates such as phosphoserine, phosphorylated sugar alcohols (mannitol and sorbitol), nucleotide monophosphate, glycerol phosphate, dihydroxyacetone phosphate and more. Substrate specificity prediction for phosphatases is notoriously difficult owing the wide variation in the structure of the “cap” domain likely responsible for substrate specificity [20]. To determine if GlcC could potentially use G6P as substrate we compared its putative structure to that of the characterized HAD C1-phosphatases. GlcC clustered with closely related enzymes from *Rouxiella* sp., *Serratia* sp. and *Francisella* sp. (Appendix A). Although C1-phosphatases displaying activity on G6P from the Huang dataset are dominated by examples from the Pseudomonadota, these enzymes are not limited to this phylum and there appears to be little structural conservation that would enable prediction of activity on G6P as enzymes that do, and do not display activity on this substrate are evenly distributed throughout the tree.

The source of glucose that forms the scaffold for attachment of 3-hydroxy fatty acids is not immediately obvious. Anyone of several activated glucose derivatives involved in glycolysis or cell wall synthesis (e.g., glucose-6-phosphate, glucose-1-phosphate, glucosamine-6-phosphate, N-acetylglucosamine-6-phosphate, UDP-glucose or UDP-N-acetylglucosamine) could be the source for the glucose moiety in the glucoselipids. However, complex nucleotide (e.g., UDP, TDP, or GDP)-activated sugars are generally less favorable as glucose donors for recent glucoselipids biosynthesis. Typically, in the case of glycolipids biosynthesis, only glycosyltransferases can catalyze the transfer of a carbohydrate moiety from these activated sugar forms to a lipid backbone such as rhamnosyltransferase RhlB in rhamnolipid biosynthesis, UDP-glucosyltransferase UGTA1 in sophorolipid biosynthesis, and mannosyltransferase EMT1P in mannosylerythritol lipids (MELs) biosynthesis [51,52,53].

### 3.2. Structural Characterization of Produced Glucoselipids

A total of 384 mg and 263 mg of crude extracts were obtained after performing glucoselipids extraction on 500 mL of cell-free supernatant from the bioreactor cultivation of *E. coli* pCAT2 and pAFP1, respectively. The crude extracts were then subjected to individual purification step using MPLC and each fraction was visualized using HPTLC and subsequent staining. Fractions (46–51) containing glucosedilipids from *E. coli* pCAT2 began to elute at approximately 65% acetonitrile gradient while fractions (37–40) containing glucosemonolipids from *E. coli* pAFP1 eluted at approximately 40% acetonitrile gradient, suggesting more hydrophilic nature of glucosemonolipids. A final amount of 69 mg purified glucosedilipids in the form of white amorphous substance and 35 mg purified glucosemonolipids in the form of pale-yellow amorphous substance were obtained at the end of purification step. To gain a clearer understanding of the chemical structure of the synthesized glucoselipids, the purification procedures were repeated. The first purification was performed to collect and analyze specific fractions using LC-ESI-MS/MS, in order to study the variation in congeners and calculate their relative amounts. The process was repeated to isolate a targeted congener in a sufficient quantity for detailed structural analysis using NMR. For quality assurance regarding sample purity, fraction 48 (from *E. coli* pCAT2) was dried and subjected to NMR analysis, while fraction 37 through 40 (from *E. coli* pAFP1) were individually dried and analyzed using NMR.

Evaluation of the 1D and 2D NMR spectra of fraction 48 and comparison with the ^1^H and ^13^C NMR data of glucosedilipid from *R. badensis* DSM 100043^T^ unambiguously established its identical chemical structure (Figure 3 and Appendix A) [10]. A distinct peak at 4.65 ppm in fraction 48 (*E. coli* pCAT2), which is entirely absent in glucosedilipid sample from *R. badensis* DSM 100043^T^, is attributed to traces of water present in the sample. While analysis of 1D and 2D NMR spectra including COSY, TOCSY, selTOCSY, HSQCTOCSY, HSQC and HMBC of fraction 37 from *E. coli* pAFP1 revealed a mixture of three pairs of glucosemonolipid of *α*- and *β*-anomeric glucopyranosyl moieties substituted with one fatty acid each (Figure 4 and Appendix A). The structure of the latter was unambiguously established as 3-hydroxydecanoic acid (C_10:0_) by evaluation of the 2D NMR spectra. The acylation positions of the two major glucosemonolipid pairs were determined by HMBC of the respective sugar proton with the carboxyl C of the fatty acid. Thus, a HMBC correlation between 2-H*α*/*β* at δ 4.64/4.72 ppm and carboxyl carbon C-1′ at δ 173.14/172.70 ppm indicated the C-2 monosubstituted glucopyranose as major compound **1** in fraction 37 whereas a HMBC correlation between 3-H*α*/*β* at δ 5.28/4.98 ppm and carboxyl carbon C-1′ at δ 173.38/173.63 ppm showed the C-3 monosubstituted glucopyranose as second compound **2** in fraction 37. The C-6 acylation side of the third monosubstituted *α*- and *β*-glucopyranosyl pair **3** was only present in trace amounts and tentatively deduced by, e.g., the low-field-shifted methylene 6-H_a/b_ (*β*-anomer) at δ 4.46 and 4.23 ppm compared to methylene 6-H_a/b_ (*β*-anomer) at δ 3.91 and 3.71 ppm of compound **1** (Appendix A). ^1^H signal integration yielded a 59:28:13 ratio (**1**:**2**:**3**) of the three glucosemonolipid Glu-C_10:0_ pairs **1**–**3**. Although three additional MPLC fractions of *E. coli* pAFP1 were isolated (fractions 38–40), their purity was insufficient for full structural elucidation using NMR due to significant spectral overlaps. Therefore, MS-guided structural analysis was then performed to characterize all purified fractions instead.

To further investigate the structure and congener composition of the produced glucoselipids, purified products from both strains were thoroughly examined using LC-ESI-MS/MS. The base peak chromatogram in the negative ion mode exhibited a prominent peak at RT 13.26 min for *E. coli* pCAT2’s purified product (pooled fraction 46–51) as shown in Figure 5A. The mass spectrum at corresponding retention time showed a deprotonated molecular ion [M-H]^−^ with a *m*/*z* of 545.3335 and formic acid adduct [M+FA-H]^−^ with a *m*/*z* of 591.3393. These findings enabled the determination of the molecular formula C_28_H_49_O_10_ (error: 0.59 ppm). Furthermore, the mass spectrum exhibited two in-source fragment ions, which were likewise observed as prominent signals in the negative-ion-mode MS/MS spectrum of *m*/z 545.3335 (Figure 5B). These fragment ions corresponded to the fatty acid components 3-hydroxydecanoic acid (C_10:0_) and 3-hydroxy-5-dodecenoic acid (C_12:1_) with *m*/z of 187.1333 and 213.1492, respectively. Therefore, the compound was designated as glucosedilipid Glu-C_10:0_-C_12:1_ informed by similarity with MS data of *R. badensis* DSM 100043^T^ glucosedilipid previously reported [10]. The acylation positions of C_10:0_ and C_12:1_ at C-3 and the C-2 position of the glucopyranosyl moiety were also unambiguously determined from ^13^C HMBC spectra of NMR results. Therefore, the structure of glucosedilipid Glu-C_10:0_-C_12:1_ elucidated by comprehensive NMR analysis was fully confirmed by the mass spectrometry results obtained from high-resolution LC-ESI-MS/MS. This findings suggest that the glucosedilipid operon was successfully expressed heterologously by *E. coli* pCAT2.

Moreover, Figure 5A and Table 4 demonstrate that *E. coli* pCAT2 predominantly produced Glu-C_10:0_-C_12:1_ (91.04%) while the minor glucosedilipid congeners Glu-C_10:0_-C_11:0_ (4.54%) and Glu-C_10:0_-C_12:0_ (4.42%) were present in smaller amounts. The relative abundance of each congener was calculated based on the corresponding peak areas obtained from the extracted ion chromatogram (XIC) of the respective congener (Appendix A). Following a thorough analysis of the minor congeners, the mass spectrum at RT 14.79 min exhibited a deprotonated ion [M-H]^−^ at *m*/*z* 547.3492, indicative of a molecular formula of C_28_H_51_O_10_, as depicted in Appendix A. The MS/MS spectrum obtained from the fragmentation of the parent ion [M-H]^−^ with *m*/*z* 547.3492 revealed two intense signals consistent with fatty acids, with m/z 187.1336 (C10:0) and 215.1649, respectively. The *m*/*z* value of 215.1649 can be calculated as the difference of 2 (the addition of two hydrogen atoms) from the C_12:1_ fatty acid. This results in the designation of 3-hydroxy-5-lauric acid (C_12:0_). Subsequent analysis of the congener compound yielded the following structure: Glu-C_10:0_-C_12:0_. An additional minor congener was detected based on the mass spectrum at RT 12.80 min where it exhibited a deprotonated ion [M-H]^−^ with *m*/*z* 533.3331 and a predicted molecular formula of C_27_H_49_O_10_ (error: 1.29 mmu) (Appendix A). MS/MS data from the parent ion also exhibited two intense typical fatty acid signals with *m*/*z* 187.1333 (C_10:0_) and 201.1491. The *m*/z value of 201.1491 can be explained as Δ*m*/*z* of −14 (reduction of one carbon chain -CH2-) to the C_12:0_ fatty acid. Accordingly, the fatty acid is designated as 3-hydroxy-5-undecanoic acid (C_11:0_) and the congener compound was deduced as Glu-C_10:0_-C_11:0_. Notably, all glucosedilipid congeners produced by *E. coli* pCAT2 contained C_10:0_ fatty acid paired with a second distinct fatty acid. In addition, the XICs from all glucosedilipid congeners revealed the presence of both minor and major isomers in each congener (Appendix A). A comparison of the ^1^D selective TOCSYs NMR with those of the XICs of each congener indicated that the major and minor isomers were glucosedilipids with *α*- and *β*-anomeric glucopyranosyl moieties, respectively [54].

LC-ESI-MS/MS analysis of *E. coli* pAFP1’s purified product (pooled fraction 37–40) demonstrates that it synthesizes glucosemonolipid congeners, with Glu-C_10:0_ (33.95%) and Glu-C_12:1_ (41.44%) being the two most prevalent congeners, which comprise the primary components of the glucosedilipid Glu-C_10:0_-C_12:1_ (Figure 6A). As demonstrated in Table 4, the analysis identified additional minor congeners, including Glu-C_12:0_ (4.07%) and Glu-C_11:0_ (3.63%), which have also been observed to contribute to the formation of minor glucosedilipid congeners. Interestingly, an additional congener Glu-C_14:0_ which had not been previously detected was also identified with a relatively high abundance of 16.91%. The relative abundance of each congener was calculated based on the corresponding peak areas obtained from the XIC of the respective glucosemonolipid congener (Appendix A). As illustrated in Figure 6A, the base peak chromatogram in the negative ion mode demonstrates that the glucosemonolipid Glu-C_10:0_ elutes within the RT interval of approximately 6.3–7.1 min as three isomeric forms. This finding is consistent with the NMR results, which revealed the presence of glucosemonolipid Glu-C_10_ in three distinct compounds differing in their acylation positions. The mass spectrum at RT 6.34 min exhibited the deprotonated molecular ion [M-H]^−^ with *m*/*z* 349.1869 and a predicted molecular formula of C_16_H_29_O_8_ (calcd: 349.1868; error: 0.03 ppm) (Figure 6B). The elimination of the glucosyl moiety (162 Da) results in the production of a remnant of *m*/*z* 187.1332, which was also identified in the mass spectrum as a prominent signal, commonly associated with 3-hydroxydecanoic acid (C_10:0_). The mass spectra (ESI full MS and MS^2^) of the other glucosemonolipid congeners produced by *E. coli* pAFP1, along with their corresponding fragmentation patterns, are presented in Appendix A.

Notably, one of the minor glucoselipid congeners produced by the *E. coli* pCAT2 and pAFP1 contained an odd-chain hydroxylated C_11:0_ fatty acid. This finding is of particular interest, as *E. coli* typically synthesizes even-chain fatty acids, and the occurrence of an odd-chain fatty acids is rare [55]. It is possible that this observation can be explained by several factors, including metabolic alterations resulting from the heterologous gene expression or the formation of artifacts during the downstream processing and sample preparation steps.

In view of the previously outlined structure of glucosedilipid and the subsequent validation in the present study, a salient question emerges: by what mechanism does the activity of these three identified genes culminate in the biosynthesis of glucosedilipid [10]? The expression of distinct gene combinations has revealed that the phosphatase GlcC is indispensable for the synthesis of compounds, indicating that, within the context of *Escherichia coli*, no other gene can adequately substitute for this function. Alternatively, if such a substitution were possible, the resulting compound production would be below our detection limit. This finding further substantiates the sequential mechanism of the acetyltransferase GlcA, which functions by initially catalyzing acylation at the C-2 hydroxyl group of the glucose scaffold with a 3-hydroxy fatty acid. Subsequent to this initial acylation, GlcA facilitates the addition of another 3-hydroxy fatty acid to the hydroxyl group at the C-3 position by GlcB, thereby culminating in the completion of glucosedilipid synthesis. This finding indicates that the acyltransferase GlcB recognizes the glucosemonolipids as a substrate and is unable to catalyze direct acylation of the activated glucose scaffold. Nevertheless, the elucidation of the complete biosynthetic pathway of the recently identified glucosedilipid would be a particularly intriguing direction for future research.

### 3.3. Fed-Batch Bioreactor Cultivation of E. coli pCAT2 for Glucosedilipid Production

Fed-batch cultivations of *E. coli* pCAT2 were carried out in a pilot-scale bioreactor system to compare recombinant glucosedilipid production performance with the previously described bioreactor cultivation using the wild-type producer *R. badensis* DSM 100043^T^ [10]. As can be seen in Figure 7, initial glucose concentration of 25 g/L was completely consumed after 10 h cultivation during the batch phase yielding an average CDW of 8.10 ± 0.25 g/L and relatively low glucosedilipid concentration of 0.21 ± 0.03 g/L. Subsequently, the feeding phase was started with the exponential pumping of feed solution and IPTG injection, maintaining a fixed growth rate of 0.2 h^−1^. The feeding phase lasted for 15 h, thus giving a total cultivation time of 25 h.

With fed-batch process utilizing glucose as the sole carbon source, high cell density was achieved with maximum cell dry weight of 59.5 ± 1.3 g/L. There was no glucose accumulation detected during the feeding phase at a set growth rate of 0.2 h^−1^. As can be seen in Figure 7, glucosedilipid production followed the trend of biomass formation where 2.34 ± 0.04 g/L glucosedilipid concentration was reached at the end of cultivation. The glucosedilipid titer achieved in this study is 55-fold higher than that produced by the wild-type producer *R. badensis* DSM 100043^T^ in batch bioreactor cultivation as previously described by Harahap et al., 2025 [10]. The stable product accumulation across the cultivation suggests that the glucosedilipid is not rapidly consumed or broken down by *E. coli*. This stability enhances further downstream processing predictability and simplifies product recovery.

## 4. Conclusions

Through the function-based screening of the *R. badensis* DSM 100043^T^ genome library, we identified an operon containing three genes responsible for the biosynthesis of glucosedilipid confirmed by mass spectrometry and NMR-based structure elucidation. The genes were identified to encode N-acetyltransferase GlcA (ORF1), acyltransferase GlcB (ORF2), and phosphatase/HAD GlcC (ORF3). Heterologous expression of this operon (pCAT2) in *E. coli* resulted in the production of identical *R. badensis* DSM 100043^T^ glucosedilipid Glu-C_10:0_-C_12:1_ as its major congener. In contrast, deletion of acyltransferase-encoding gene *glc*B (ORF2) in *E. coli* pAFP1 led to the production of glucosemonolipid, particularly Glu-C_10:0_ and Glu-C_12:0_ as its primary congeners, with a predominant acylation site at the C-2 position of the glucose moiety. This finding suggests that GlcA functions sequentially by first catalyzing the acylation, preferably at the C-2 hydroxyl group of the glucose scaffold, using a 3-hydroxy fatty acid as the acyl donor. These findings also indicate that the acyltransferase GlcB likely catalyzed the second acylation of 3-hydroxy fatty acid molecule to the glucose moiety, thereby providing a basis for subsequent enzyme characterization studies. Furthermore, the fed-batch high cell density fermentation of *E. coli* pCAT2 utilizing glucose as the sole carbon source resulted in 2.34 g/L glucosedilipid titer after 25 h fermentation, which was 55-fold greater than the previously reported titer in the batch bioreactor fermentation of *R. badensis* DSM 100043^T^ using glycerol as the carbon source. The results presented lay the foundation for future studies aimed at elucidating the biosynthesis pathway and enabling the production of glucoselipids for biotechnological applications as novel biosurfactants.

## Figures and Tables

**Figure 1 microorganisms-13-01664-f001:**
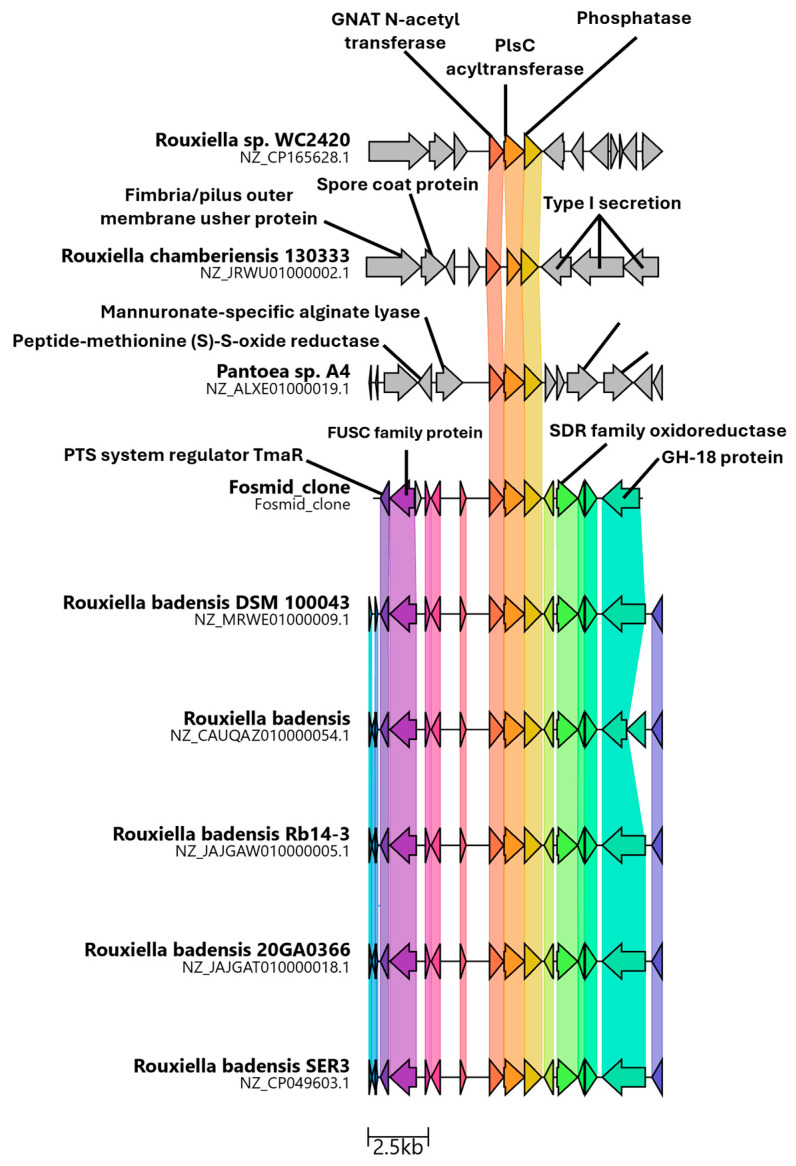
Comparison of the genomic regions encoding the glucoselipid ORFs found in *Rouxiella* sp. and closely related pathways.

**Figure 2 microorganisms-13-01664-f002:**
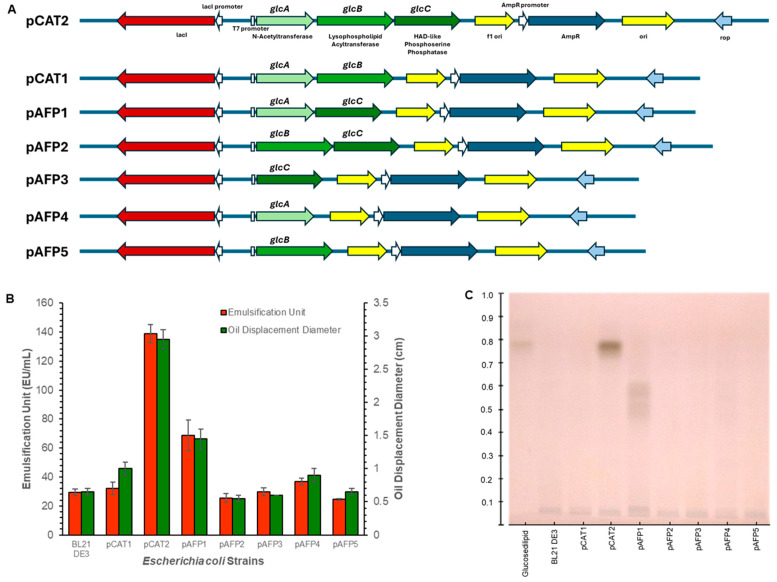
Effect of gene permutation on different expression strains’ phenotypic characteristics: Schematic diagram of different plasmid constructs carrying different ORF(s) combinations, where ORF 1 is *glc*A, ORF2 is *glc*B, and ORF3 is *glc*C (**A**); Emulsification unit and oil displacement diameter of supernatant from all expression strains (**B)**; and p-anisaldehyde-stained RP18 TLC plate of crude extracts showing prominent bands at *R_f_* 0.79 and *R_f_* 0.5–0.6 for *E. coli* pCAT2 and pAFP1, respectively (**C**). Please note in subfigure (**C**) that the glucosedilipid on the first lane was obtained from *R. badensis* DSM 100043^T^ while all other lanes were sample from *E. coli* expression strains.

**Figure 3 microorganisms-13-01664-f003:**
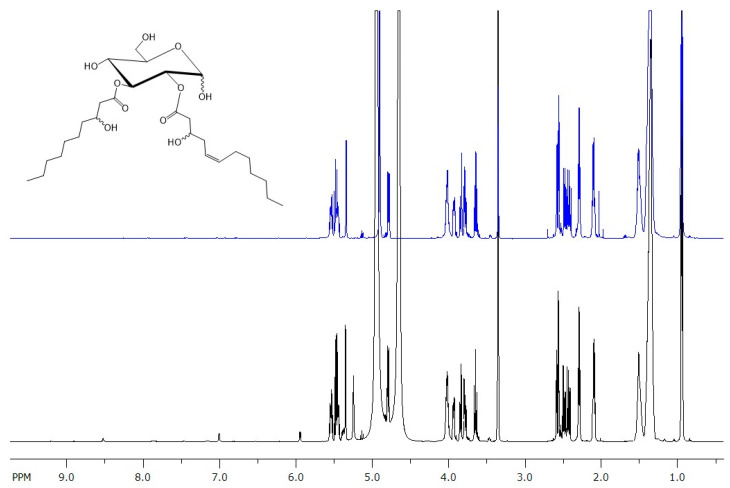
Comparison of ^1^H NMR spectra of freshly dissolved glucosedilipid from *R. badensis* DSM 100043^T^ (blue) and *E. coli* pCAT2 (black) confirming the identical structure of both products as Glu-C_10:0_-C_12:1_ [10]. Some impurities of unknown structure were detected between δ 5.80 and 8.80 ppm in the spectrum of glucosedilipid from fraction 48 (*E. coli* pCAT2).

**Figure 4 microorganisms-13-01664-f004:**
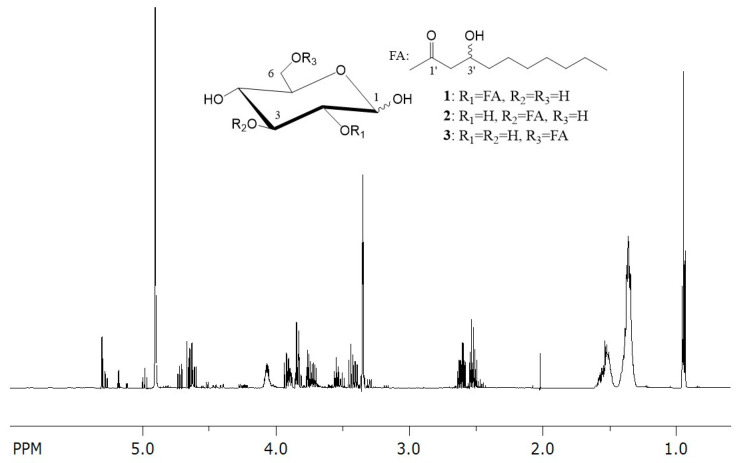
^1^H NMR spectrum of fraction 37 and the chemical structure of glucosemonolipids Glu-C_10:0_ **1**–**3** identified in fraction 37.

**Figure 5 microorganisms-13-01664-f005:**
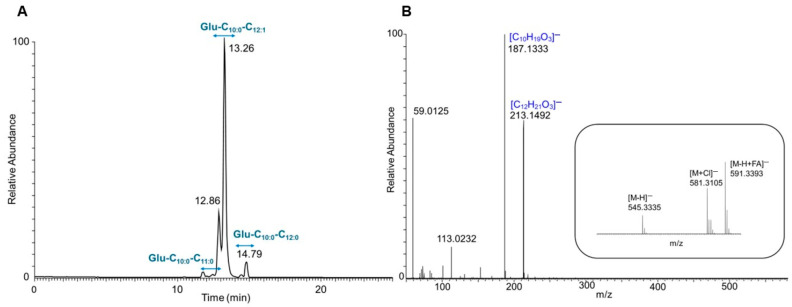
LC-ESI-MS/MS analysis of the purified glucosedilipids produced by *E. coli* pCAT2 (pooled fraction 46–51). The total ion chromatogram (TIC) in the negative ion mode (**A**) shows diverse glucosedilipid congeners including their isomers. The mass spectrum (**B**, inset) in the negative ion mode at retention time of 13.26 min shows Glu-C_10:0_-C_12:1_ as the most abundant congener. The MS/MS spectrum of the protonated molecular ion *m*/*z* 545.3335 (**B**) exhibits a high degree of similarity to the MS/MS spectra of *R. badensis* DSM 100043^T^ glucosedilipid [10].

**Figure 6 microorganisms-13-01664-f006:**
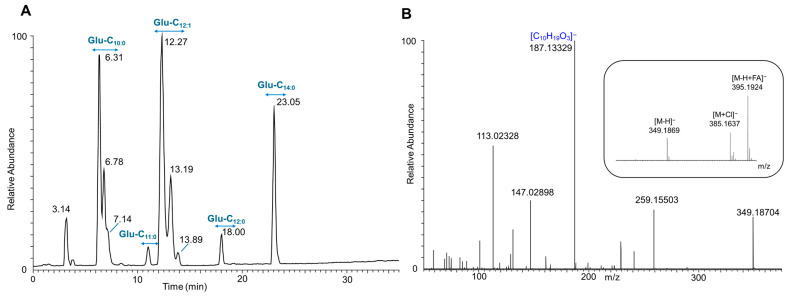
LC-ESI-MS/MS analysis of the purified glucosemonolipids produced by *E. coli* pAFP1 (pooled fraction 37–40). The total ion chromatogram (TIC) in the negative ion mode (**A**) shows diverse glucosemonolipid congeners including their isomers. The mass spectrum (**B**, inset) in the negative ion mode at retention time of 6.31 min exhibited a deprotonated molecular ion with a *m*/*z* 349.1869. The MS/MS spectrum of the deprotonated molecular ion *m*/*z* 349.1869 (**B**) revealed fragmentation patterns of Glu-C_10:0_ which is one of the most abundant glucosemonolipid congeners. This observation is consistent with the results obtained by nuclear magnetic resonance (NMR) analysis.

**Figure 7 microorganisms-13-01664-f007:**
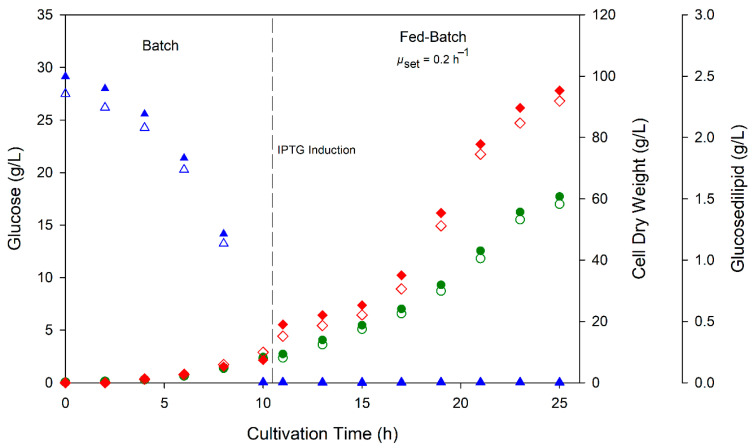
Time course of fed-batch bioreactor cultivation for the recombinant production of glucosedilipid by *E. coli* pCAT2 showing CDW (green circle), glucose concentration (blue triangle) and glucosedilipid concentration (red diamond) from biological duplicate experiments.

**Table 1 microorganisms-13-01664-t001:** List of strains and plasmids used in this study.

Name	Genotype	Reference
**Strains**		
*Rouxiella* badensis DSM 100043^T^	Wild-type	[8]
*Escherichia coli* BL21(DE3)	*E. coli* BL21 F^-^ *ompT hsdS*(*rB* – *mB* –) *dcm* + *Tetr gal λ*(*DE3*) *endA Hte* [*argU proL Camr*] [*argU ileY leuW Strep*/*Specr*]	Agilent Technologies (Santa Clara, CA, USA)
EPI300	F^−^ *mcrA* Δ(*mrr*-*hsdRMS*-*mcrBC*) *φ80dlacZΔM15* Δ*lacX74 recA1 endA1 araD139* Δ(*ara*, *leu*)*7697 galU galK λ^−^ rpsL nupG trfA tonA dhfr*	Epicentre (Illumina, San Diego, CA, USA)
K12 JM109		[15]
*Pseudomonas putida* MBD1	*P. putida* KT2440 derivative; Kan*^r^*; *φC31 attB site*^+^	[14]
54F3	*P. putida* MBD1 containing 54F3 fosmid for expression of lyso-ornithine lipid biosurfactant	[16]
**Plasmids**		
pCCERI	pCC1FOS derivative, Cm^r^, Ap^r^, φC31 integrase, *attP*	[17]
pER1.3.50.2	pRK2013 derivative with *trfA* gene deleted, Kan^r^	[17]
Fosmid 1.8.H6	pCCFOS1 clone containing 2,399,010 bp to 2,429,311 bp of *R. badensis* (CP114060)	This study
pET21a(+)	Expression vector with a C-terminal His-tag, Amp*^r^*, *ori*, T7 promoter and terminator, MCS	Novagen (Merck KGaA, Darmstadt, Germany)
pCAT2	pET21a(+) derivative; ORF 1; ORF2; ORF3	This study
pCAT1	pET21a(+) derivative; ORF 1; ORF2	This study
pAFP1	pET21a(+) derivative; ORF 1; ORF3	This study
pAFP2	pET21a(+) derivative; ORF 2; ORF3	This study
pAFP3	pET21a(+) derivative; ORF3	This study
pAFP4	pET21a(+) derivative; ORF1	This study
pAFP5	pET21a(+) derivative; ORF 2	This study

**Table 2 microorganisms-13-01664-t002:** LC-ESI/MS settings for analysis of purified glucoselipid samples.

Parameters	*E. coli* pCAT2 Glucoselipids	*E. coli* pAFP1 Glucoselipids
Column temperature	40 °C	50 °C
Injection volume	0.37 µL	3.0 µL
Mobile phase	A: 0.2% formic acid in water B: 0.2% formic acid in acetonitrile	A: 0.2% formic acid in water B: 0.2% formic acid in methanol
Gradient elution	50–53% B from 0 to 15 min, 53–67% B from 15 to 17 min, 67–75% B from 17 to 25 min, 75–90% B from 25 to 30 min, 90–90% B (isocratic) from 30 to 35 min, 90–50% B from 35 to 36 min, 50–50% B from 36 to 37 min	45–53% B from 0 to 5 min, 53–59% B from 5 to 10 min, 59–90% B from 10 to 20 min, 90–90% B (isocratic) from 20 to 25 min, 90–45% B from 25 to 26 min
Scan range	200–1500 *m*/*z*	100–1400 *m*/*z*
(N)CE *	32	10

* Normalized collision energy.

**Table 3 microorganisms-13-01664-t003:** The top two structural similarity hits using Foldseek for GlcA, GlcB and GlcC against AFDB50 and PDB100.

Protein Description	Organism	Prob.	E Value	Seq. Ident %	Accession
**GlcA**					
N-acetyltransferase domain-containing protein	*Pseudomonas fluorescens*	1.00	8.92 × 10^−23^	40.4	AF-A0A5E6RK03-F1-model_v4
Crystal structure of a GNAT superfamily PA3944 acetyltransferase in complex with CoA	*Pseudomonas aeruginosa* PAO1	1.00	6.33 × 10^−12^	22	6EDD
**GlcB**					
Uncharacterized protein	*Rouxiella badensis*	1.00	1.90 × 10^−53^	100	AF-A0A1X0WHA5-F1-model_v4
Crystal structure of the 1-acyl-sn-glycerophosphate (LPA) acyltransferase, PlsC	*Thermotoga maritima* MSB8	1.00	9.26 × 10^−6^	14.7	5KYM
**GlcC**					
Uncharacterized protein	*Serratia* sp. M24T3	1.00	1.18 × 10^−39^	77.8	AF-I0QXG7-F1-model_v4
Crystal structure of a phosphoserine phosphohydrolase-like protein	*Francisella tularensis* SCHU S4	1.00	7.70 × 10^−13^	23.4	3KD3

**Table 4 microorganisms-13-01664-t004:** Summary of the glucoselipids congeners produced by strains *E. coli* pCAT2 and *E. coli* pAFP1 as identified by means of LC-ESI-MS/MS analysis.

Strain	Glucoselipid	Molecular Formula	RT * [min]	*m*/*z* [M-H]^−^	Fatty Acid’s *m*/*z* [M-H]^−^		Relative Abundance (%)
***E. coli* pCAT2**	Glu-C_10:0_-C_11:0_	C_27_H_50_O_10_	12.3	533.333	C_11_H_21_O_3_^−^ (201)	C_10_H_19_O_3_^−^ (187)	4.54
			12.8				
	Glu-C_10:0_-C_12:1_	C_28_H_50_O_10_	12.8	545.333	C_12_H_21_O_3_^−^ (213)	C_10_H_19_O_3_^−^ (187)	91.04
			13.2				
	Glu-C_10:0_-C_12:0_	C_28_H_52_O_10_	14.4	547.349	C_12_H_23_O_3_^−^ (215)	C_10_H_19_O_3_^−^ (187)	4.42
			14.7				
***E. coli* pAFP1**	Glu-C_10:0_	C_16_H_30_O_8_	6.3	349.186	C_10_H_19_O_3_^−^ (187)	-	33.95
			6.7				
			7.1				
	Glu-C_11:0_	C_17_H_32_O_8_	11.0	363.202	C_11_H_21_O_3_^−^ (201)	-	3.63
			11.8				
	Glu-C_12:0_	C_18_H_34_O_8_	18.0	377.218	C_12_H_23_O_3_^−^ (215)	-	4.07
			18.4				
	Glu-C_12:1_	C_18_H_32_O_8_	12.3	375.202	C_12_H_21_O_3_^−^ (213)	-	41.44
			13.1				
			13.8				
	Glu-C_14:0_	C_20_H_38_O_8_	23.0	405.249	C_14_H_27_O_3_^−^ (243)	-	16.91

* Retention times are provided for each identified isomer.

## Data Availability

The original contributions presented in this study are included in the article/Appendix A. Further inquiries can be directed to the corresponding author.

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
