# Peer review of "Glucoselipid Biosurfactant Biosynthesis Operon of Rouxiella badensis DSM 100043T: Screening, Identification, and Heterologous Expression in Escherichia coli"

_microorganisms, 2025, doi:10.3390/microorganisms13071664_

Round 1

Reviewer 1 Report

Comments and Suggestions for Authors

Glucoselipid biosurfactants exhibit excellent surface-active properties and stability, making them highly promising for applications in various fields.

Rouxiella badensis has been proven to produce a novel glucoselipid biosurfactant. The authors identified biosynthetic genes for glucoselipid using a function-based screening method and confirmed by heterologous expression in E. coli. The expect products have been analysed by HPLC and MS/MS, and the structures were further elucidate by NMR. Finally, the fed-batch bioreactor cultivation of recombination E. coli yielded a maximum glucosedilipid titer which is 55-fold higher than R. badensis wild type using glucose as carbon source.

This work successfully identifies the biosynthetic operon for novel glucoselipid biosurfactants from R. badensis and provides new insights into the biosynthesis and application of glucoselipid biosurfactants.

(1) The results is not very logical. The authors combined the bioinformatics analysis and gene expression in E. coli make the first part of the result more complicated. When studying the impact of gene arrangement on glucoselipid biosynthesis, although several recombinant E. coli strains with different gene combinations were constructed, there is a lack of control experiments examining the expression of each gene individually or in combination with other unrelated genes. This makes it difficult to accurately assess the independent role and specificity of each gene in glucoselipid biosynthesis.

(2) The figures in the main text need to be improved. Eg. The constructs in figure

(3) How to quantify the yield of glucoselipid? Is there any standard curve or using standards?

(4) The genes in this work should get the accession number and no need to list the amino acid and nucleotide sequence in table S2.   

(5) The structures predicted using Alphafold does not seem to contribute positively to the results.

(6) Some grammatical errors and inaccuracies in word usage that need to be paid more attention to enhance the clarity of the expression. Eg. “m/z should be italic.

Reviewer 2 Report

Comments and Suggestions for Authors

The manuscript entitled: “Glucoselipid Biosurfactant Biosynthesis Operon of Rouxiella badensis DSM 100043T: Screening, Identification, and Heterologous Expression in Escherichia coli”. The manuscript is extremely well written, clear and easy to follow. In my opinion, this work is highly relevant due to the vital shift towards biotechnologically obtained surfactants.

I just have three questions/suggestions, and a few minor amendments:

  1. The Conclusion section, in my opinion, falls short of what a Conclusion should include, namely in this case, the envisaged potential for scaling up, including its foreseen advantages (for instance the use of glycerol that should be underscored) and/or limitations. Including an brief economical overview in comparison to chemically obtained surfactants.
  2. What is the performance of the biosurfactant? I am sorry if I missed it, but I did not observe the characterization of the surfactant, in terms of its activity.
  3. Line 187, “A mist of paraffin was sprayed on the colonies using an airbrush”. We use airbrushes in our lab, and I am not understanding how this mist was obtained considering the particularities of paraffin (that we also use for other purposes). A more detailed description of the process would be most welcomed. For instance, what is the temperature used for this process? Were the aseptic conditions maintained? How? Furthermore, a picture detailing the oil displacement diameter would be excellent.

Minor amendments:

Line 117, 140, 215, please convert rpm to gravitational force, its unit is an italicized g.

Line 243, 249, please ensure that all species names are italicized.

Line 272, please define the percentage: (w/v), (v/v)?

Line 287, the unit × g is incorrect, please use an italicized g.

Line 293, I beg your pardon, but I failed to understand the following proportion: 100:10:1, v/v/v, can the authors please clarify? For instance, the authors used 100 mL of isopropyl acetate, 10 mL of methanol and 1 mL of acetic acid?

Line 315, please avoid the use of “&”.

Line 346, the unit is missing.

Line 347, please revise “3.0 × 10E6”

Reviewer 3 Report

Comments and Suggestions for Authors

In their manuscript Harahap et al., present the identification and heterologous expression of an operon involved in glucoselipid biosynthesis. The research is well conducted and the manuscript is written in a clear and understandable way. Nevertheless I have a few major and minor comments that should be addressed prior to publication.

Major comments:

Line 149: The term “twofold extraction” occurs several times in the manuscript. However, it is not explained. Considering the context in which it appears, it could also be that there are different “twofold extractions”. Please explain and clarify.

Figure 3: I agree that both spectra look nearly identical. However, there is a very large peak at ~4.8 ppm in the E. coli sample that is completely missing in the Rouxiella sample. This peak should be explained.

Line 692: This statement is not supported by any presented data. Please provide an appropriate figure.

Minor comments:

Line 34: I think to reduce misunderstandings it should be “the acetyltransferase GlcA (ORF1)” instead of “GlcA acetyltransferase (ORF1)”. The same is true for the other two genes.

Line 60: Please consider “glycolipid production” instead of “glycolipids production”.

Line 66: I think there was only one “strain” and not “strains” assembled.

Line 97: The amount of agar should be placed directly behind the term “agar”.

Line 99: The name of plasmid pER1.3.50.2 is the end of the sentence. I think the following “and” can be removed.

Line 110: The procedure for extracting chromosomal DNA seems extremely time-consuming to me. Typical protocols for chr. DNA extraction (with or without kit) take only a few hours. Is there a specific reason why this extremely lengthy protocol lasting several days had to be used here?

Line 117: Was the centrifugation really carried out at 40°C? Please comment.

Line 208: I don’t understand why a stop codon has to be removed. If the stop codon of an ORF is removed translation cannot be terminated. Please explain.

Line 233: If there is more than one ORF it should be ORFs.

Figure 2A: The arrows representing the inserted ORFs point in the wrong direction. It would also be better if the linear plasmid map is positioned in a way that the inserted genes are in the center of the image and not at the edge.

Figure 2C: It should be indicated in the figure that the sample in the lane “Glucoselipide” was obtained from Rouxiella while all other samples were obtained from E. coli.

Line 524: How should repetition of the purification increase understanding of the chemical structure of a compound? Please explain.

Line 526: All fractions to be analyzed were “individually dried and analyzed by NMR” I guess. Thus, it is not necessary to state this twice.

Figure 641: I would strongly advise against using your own inability to clean a bioreactor properly as an explanation for the occurrence of a fatty acid. The other two explanations are perfectly adequate in my opinion.

Line 707: “hypothetically suggests” is kind of saying the same thing twice. Thus, “hypothetically” can be removed.

Round 2

Reviewer 3 Report

Comments and Suggestions for Authors

The alterations made by the authors has improved the manuscript. Nevertheless, I have some minor suggestions that should be considered.

  • The explanation of what's meant by twofold extraction is helpful. However, I find the term “twofold extraction” unfortunate as it is an extraction that was carried out twice. Thus, it might be considered to refer to the extraction as what it is, namely “was extracted twice”.
  • In Fig. 7, only the total amount of glycolipid is considered. This is the sum of all processes (synthesis and degradation) that take place in the cells. If degradation has not been investigated, degradation cannot be ruled out. The increasing proportion of glycolipid only indicates that synthesis is stronger than possible degradation. A complete lack of degradation is pure speculation that cannot be substantiated by results. Therefore, the whole aspect should be presented as it can be proven, namely:

“The stable product accumulation across the cultivation suggest that glucosedilipid is not rapidly consumed or broken down by E. coli.”

  • Line 97: The term LB should refer to both (LB) medium (liquid) and (LB) agar (solid). To avoid misunderstandings, the quantity of agar should therefore be given in brackets after the word agar. "LB medium and agar (15 g/L)"
  • The explanation “To fuse the protein with the His-tag from the pET21a vector the stop codon was removed” should also be included in the manuscript. Otherwise it would still be unclear for the reader why the stop codon was removed.
  • Similarly, the explanation that the first purification was done with pooled fractions while the second one was done with individual fractions should be mentioned. This explanation is required to understand that new insights on the chemical structure of the compound were generated by repetition of the purification.
